# Superficial Endometriosis at Ultrasound Examination—A Diagnostic Criteria Proposal

**DOI:** 10.3390/diagnostics13111876

**Published:** 2023-05-27

**Authors:** Marcelo Pedrassani, Stefano Guerriero, María Ángela Pascual, Silvia Ajossa, Betlem Graupera, Mariachiara Pagliuca, Sérgio Podgaec, Esdras Camargos, Ygor Vieira de Oliveira, Juan Luis Alcázar

**Affiliations:** 1CLINUS Ultrasound Clinic, Florianópolis 88015-200, Brazil; marcelo.pedrassani@gmail.com; 2Department of Obstetrics and Gynecology, Hospital Maternidade Carmela Dutra and Hospital Baia Sul, Florianópolis 88015-270, Brazil; 3Centro Integrato di Procreazione Medicalmente Assistita e Diagnostica Ostetrico-Ginecologica, Azienda Ospedaliero Universitaria—Policlinico Duilio Casula, Monserrato, University of Cagliari, 09042 Cagliari, Italy; gineca.sguerriero@tiscali.it (S.G.); gineca.sajossa@tiscali.it (S.A.); mkpagliuca@gmail.com (M.P.); 4Department of Obstetrics, Gynecology, and Reproduction, Hospital Universitari Dexeus, 08028 Barcelona, Spain; marpas@dexeus.com (M.Á.P.); betgra@dexeus.com (B.G.); 5Department of Obstetrics and Gynecology, Hospital das Clínicas da Faculdade de Medicina da Universidade de São Paulo, São Paulo 05403-010, Brazil; sergiopodgaec@me.com; 6Clinic of Gynecological Surgery and Endometriosis at Hospital Maternidade Carmela Dutra and Hospital Baia Sul, Florianópolis 88015-270, Brazil; 7Hospital Maternidade Carmela Dutra, Florianópolis 88015-270, Brazil; 8Department of Obstetrics and Gynecology, Clinica Universidad de Navarra, 31008 Pamplona, Spain

**Keywords:** superficial endometriosis, ultrasound, laparoscopy, diagnosis, imaging

## Abstract

The actual prevalence of superficial endometriosis is not known. However, it is considered the most common subtype of endometriosis. The diagnosis of superficial endometriosis remains difficult. In fact, little is known about the ultrasound features of superficial endometriotic lesions. In this study, we aimed to describe the appearance of superficial endometriosis lesions at ultrasound examination, with laparoscopic and/or histologic correlation. This is a prospective study on a series of 52 women with clinical suspicion of pelvic endometriosis who underwent preoperative transvaginal ultrasound and received a confirmed diagnosis of superficial endometriosis via laparoscopy. Women with ultrasound or laparoscopic findings of deep endometriosis were not included. We observed that superficial endometriotic lesions may appear as a solitary lesions, multiple separate lesions, and cluster lesions. The lesions may exhibit the presence of hypoechogenic associated tissue, hyperechoic foci, and/or velamentous (filmy) adhesions. The lesion may be convex, protruding from the peritoneal surface, or it may appear as a concave defect in the peritoneum. Most lesions exhibited several features. We conclude that transvaginal ultrasound may be useful for diagnosing superficial endometriosis, as these lesions may exhibit different ultrasound features.

## 1. Introduction

Endometriosis is a gynecological disease characterized by the presence of endometrium-like tissue outside the uterus [1]. There are three different subtypes, namely, peritoneal endometriosis, ovarian endometriosis, and deep endometriosis [2,3].

Peritoneal endometriosis, also called superficial endometriosis, is characterized by functional ectopic endometrium-like tissue extending up to 5mm under the peritoneal pelvic surface and/or the serosa of pelvic viscera [3]. During laparoscopy, superficial endometriotic lesions appear as transparent-vesicle lesions in the very early stage, evolving to black-brown or light red-orange areas in active lesions and as 1–3 mm white fibrotic areas in inactive lesions [3,4,5]. Microscopically, for peritoneal endometriosis, some cases may exhibit the presence of endometrium-like tissue characterized by the presence of endometrial glands and stroma, or even only endometrial stroma [3]. However, these ectopic lesions are histologically diverse and rarely show menstrual cycle synchronicity with the matched eutopic endometrium [6].

The pathogenesis of endometriosis, and particularly superficial endometriosis, is not completely known. The development of endometriosis involves interacting endocrine, immunologic, proinflammatory, angiogenic, and neurogenic processes [1]. Classically, the theories of retrograde menstruation and implantation of endometrial cells and coelomic metaplasia have been advocated to explain the development of endometriosis. However, these theories do not explain the different subtypes of the clinical appearance of endometriosis and have been challenged by a recent theory of genetic-epigenetic changes postulating that a series of cumulative genetic-epigenetic incidents, related to intracellular aromatase activity resulting in intracellular estrogen production, are required for the development of endometriosis [7,8].

The actual prevalence of superficial endometriosis is not known. However, it is considered the most common subtype of endometriosis, being present in up to 80% of women suffering this disease [9]. In fact, it has been reported that in 30% of women with laparoscopically confirmed case diagnoses, it is the only type of endometriosis present [10]. Superficial endometriosis is found in 40% of asymptomatic women [8]. However, superficial endometriosis has been associated with pelvic pain and infertility [1,3,8], with an adjusted prevalence ratio of 1.83, 1.43, and 1.50 for primary infertility, severe dysmenorrhea, and deep dyspareunia, respectively, as compare to women with no endometriosis [10]. It has been reported that approximately 64% of adolescent patients receiving laparoscopy for refractory pelvic pain as an alternative to conventional medical treatment exhibited endometriosis, half of them presenting with superficial endometriosis [11].

The diagnosis of endometriosis remains difficult, as there is no reliable biomarker, and often, laparoscopy is required to confirm the presence of the disease [3,11]. This is particularly true for the case of peritoneal endometriosis [9]. In fact, current guidelines establish that laparoscopy is needed for diagnosing superficial endometriosis [12]. Certainly, imaging techniques, such as transvaginal ultrasound (TVS) and magnetic resonance image (MRI), can effectively diagnose ovarian endometriosis and deep endometriosis [13,14,15,16]. However, significant limitations have been traditionally assumed for peritoneal endometriosis [17]. Some authors have advocated the use of the so-called SonoPODgraphy, consisting of creating an acoustic window in the pouch of Douglas (POD) by the instillation of saline through the tubes, as usually performed in saline-infusion sonohysterography [18]. In this preliminary report, the accuracy of this technique in a series of 30 women with superficial endometriosis, without other types of the disease (ovarian or deep) or Douglas obliteration, was 80%.

In this paper, we aimed to describe how superficial endometriosis lesions appear at ultrasound examination, along with laparoscopic and/or histologic correlation, and propose a classification of findings.

## 2. Materials and Methods

### 2.1. Study Participants

Patients with clinical suspicion of pelvic endometriosis and no history of pelvic surgery or pelvic inflammatory disease were eligible for this study. The series was consecutive, and the women were examined from January 2021 to January 2023. Women with ultrasound findings suggesting the presence of deep endometriosis were excluded. All patients provided written signed informed consent to participate in the study. IRB approval was obtained (Institution: Maternidade Carmela Dutra; Reference: 1873304).

### 2.2. Protocol

The patients underwent a bowel preparation protocol using laxatives on the previous day, as well as a rectal enema up to two hours before the examination. The scanning protocol was in accordance with IDEA consensus [19], along with a detailed evaluation of the peritoneal surface and rectal-sigmoid anterior serosa. The presence of deep and superficial endometriotic lesions was confirmed by laparoscopic/histological correlation.

Ultrasound assessment was performed by a single expert examiner with more than 10 years of experience in sonographic evaluation of pelvic endometriosis. The equipment used was a Voluson E10 BT18 2019 (GE HEALTHCARE, Zipf, Austria) and an S10 Expert BT18 2021 (GE HEALTHCARE, Seoul, Korea) at the CLINUS Ultrasound Clinic, Florianópolis, Brazil.

### 2.3. Procedure

The transvaginal ultrasound technique used for detecting superficial endometriosis was performed with a detailed scanning of the peritoneum in the anterior, middle, posterior, and lateral compartments of the pelvis. The surface of the ovaries and serosa of the rectosigmoid were also evaluated in detail.

The ultrasound examination was performed with a volumetric endocavitary transducer RIC 5-9 MHz in both machines, using similar presets with the following parameters: frequency in HI (harmonic imaging) mode with inverted pulse harmonic, AO 100% and SRI (speckle reduction image) II 3/CR (contrast reduction). The anterior, middle, posterior, lateral, ovarian capsule, and serosa of the rectosigmoid compartments were evaluated. The image required substantial magnification for a better assessment. Ultrasound evaluation was always performed in the second phase of the menstrual cycle in order to take advantage of the presence of pelvic free fluid after ovulation. In women taking oral contraceptives and having menses, the examination was performed during menstruation. Abdominal manual evaluation to cause sliding of the structures and mainly to move the free fluid at the pouch of Douglas and towards the side compartments was always used. 

After ultrasound evaluation, all women underwent surgery performed by three different expert surgeons who were aware of ultrasound findings and specifically looked for those lesions observed at ultrasound examination. The revised American Society for Reproductive Medicine (ASRM) classification was used for staging the disease [20].

## 3. Results

In a series of 52 consecutive women, there were 68 laparoscopically and histologically lesions confirmed as superficial endometriosis. The ASRM stages in these 52 women were as follows: 45 (86.5%) stage I and seven (13.5%) stage II. No patient exhibited deep endometriosis, as determined by laparoscopic findings.

We observed that cystic lesions might appear as a solitary lesion, multiple separate lesions, lesions grouped into a linear cluster, or lesions grouped with a honeycomb appearance. A hyperechogenic focus, probably caused by hemosiderin deposit or calcification, is frequently observed inside each cystic lesion. However, it is important to observe that small cystic lesions may present specular artifacts with hyper-refringence of the wall in a half ring halo-shape that may simulate a hyperechogenic focus. Moreover, the solitary lesions had a maximum diameter of 1 to 5 mm. 

Regarding the peritoneal surface, it was observed that the lesions might be predominantly convex or concave, in the latter case due to a peritoneal retraction/defect forming a small pocket.

Another finding often associated with cystic lesions is the presence of hypoechogenic tissue reflecting stromal reaction/fibrosis. It could involve the cysts, be located only on the peritoneal face of the lesions, or entertain small cysts grouped into a honeycomb, producing a hypoechogenic spongiform aspect.

Based on the findings we observed, we propose that superficial endometriotic lesions might be classified into three main categories based on their distribution: solitary lesions (one single lesion), multiple separate lesions (more than one lesion with apparent normal peritoneum in between the lesions), and cluster lesions (more than one lesion with no apparent normal peritoneum in between the lesions) (Figure 1). Additionally, each one of these lesions may exhibit one or more particular ultrasound features, described as follows:The presence of hypoechogenic associated tissue (hypoechoic areas surrounding a small cyst area; we called this a “hat”). This tissue does not protrude or invaginate the peritoneal surface.The lesion may be convex, protruding from the peritoneal surface into the peritoneal cavity (we called this “bulging”), or it may appear as a concave defect in the peritoneum (we called this a “pocket”).The presence of hyperechoic foci (we called this a “pearl”).The presence of velamentous (filmy) adhesions associated to the lesion (we called this a “veil”).

Examples of these lesions are shown in following figures.

### 3.1. Cystic Solitary Lesion

This type of lesion may appear with different forms, including as a single echogenic lesion convex to the peritoneal surface (Figure 2).

It can also present as a single lesion convex (“bulging”) to the peritoneal surface with a hypoechogenic tissue surrounding the lesion (Figure 3 and Figure 4).

Moreover, it can present as a lesion concave to the peritoneal surface (“pocket”) (Figure 5).

Additionally, it may present as a lesion concave to the peritoneal surface (pocket), with hypoechogenic tissue surrounding the lesion (“hat”) (Figure 6).

These lesions usually do not appear with a hyperechoic foci or associate to filmy (velamentous) adhesions.

### 3.2. Cystic Multiple Separate Lesions

As stated above, the main common feature of these lesions is that they appear as dispersed lesions with apparently normal peritoneum in between them. They may appear as lesions convex to the peritoneal surface (Figure 7).

They also may appear as dispersed lesions convex to the peritoneal surface with hypoechoic tissue (hat) (Figure 8) or with a hyperechogenic foci (pearl) (Figure 9 and Figure 10).

These lesions may also appear as dispersed lesions convex to the peritoneal surface with velamentous (filmy) adhesions, without hyperechogenic foci (Figure 11), or with hyperechogenic foci (Figure 12).

Additionally, these lesions may appear as lesions concave to the peritoneal surface (pocket), with hypoechogenic tissue surrounding the lesion (Figure 13).

These lesions may appear as associated to velamentous adhesions (Figure 14 and Figure 15).

### 3.3. Cystic Lesions Arranged in a Cluster

The main common feature of these lesions is that they may appear as lesions arranged in a cluster, with no apparent normal peritoneum in between the lesions. The disposition may be linear, or as a honeycomb structure.

Figure 16 shows cystic lesions, mostly concave to the peritoneal surface (pocket), in a linear cluster disposition.

These lesions may exhibit hyperechoic foci (Figure 17, Figure 18 and Figure 19) or even hyperechoic foci with hypoechogenic tissue surrounding it (Figure 20).

Some clustered lesions may be grouped with honeycomb appearance. These lesions appear as lesions mostly convex (bulging) to the peritoneal surface (Figure 21, Figure 22, Figure 23, Figure 24 and Figure 25).

These lesions may exhibit hyperechoic foci and/or hypoechoic tissue surrounding the lesion (Figure 26 and Figure 27).

Additionally, these lesions may appear as concave to the peritoneal surface (pocket), with hypoechoic tissue surrounding the lesion, with or without a hyperechoic foci (Figure 28, Figure 29 and Figure 30).

Figure 31 schematically summarizes these examples.

In the series analyzed, we observed that superficial endometriotic lesions rarely appear as a solitary lesion with a single associated ultrasound feature. Most lesions appear as multiple or clustered lesions, with more than one ultrasound feature. Out of the 68 lesions analyzed, 34 (50%) appeared as honeycomb cluster bulging or pocket lesions, 18 (26.5%) appeared as multiple bulging or pocket lesions, 10 (14.7%) appeared as linear cluster pocket lesions, and 6 (8.8%) appeared as solitary bulging or pocket lesions.

## 4. Discussion

Approximately 80% of women who have endometriosis have superficial lesions, whereas 20% have deep endometriosis (DE) [21]. Ultrasound has been demonstrated to exhibit good diagnostic performance for detecting deep endometriosis [13,14,15,16], but scanning patients for detecting only deep endometriosis would mean that a high proportion of women with endometriosis would not be diagnosed. This is the main reason why it is so important to diagnosis superficial disease.

The 2005 ESHRE guidelines on endometriosis state that laparoscopic visualization of suspicious lesions is the “gold standard” for the definitive diagnosis of endometriosis [22]. The typical appearance of endometriosis is described as a superficial “powder-burn” or “gunshot” lesion that is black, dark-brown, or blue, but subtle lesions which are red or clear and small, or cysts with hemorrhage or white areas of fibrosis may also be signs of endometriosis [5,22]. It has been observed that white and mixed-color lesions exhibited a higher percentage of histologically confirmed endometriosis than black lesions [23]. Nisolle and Donnez [2] attributed the color of the endometriosis lesions to changes in the lesion’s age, starting out as a red lesion, then progressing to black, and finally to white [24,25,26,27]. Thus, endometriosis is not limited to a single color and may be confirmed more frequently in a multi-colored lesion. At present, it is unclear as to which color is most frequently associated with endometriosis. Furthermore, atypical (clear, grainy, red, and flame) endometriotic lesions are more common than the “classic” dark blue-black lesions in the adolescent population experiencing pelvic pain [22,23]. Martin reported a pattern of evolution of subtle lesions in adolescence to more classic disease a decade later [24]. Redwine showed that clear and red lesions occur at a mean age of 10 years earlier than do black lesions [28].

Superficial endometriosis is a relevant clinical problem because of its prevalence and diagnostic difficulties. It has been reported that superficial endometriosis may be the only entity diagnosed in up to 30% of women undergoing laparoscopy for pelvic pain [10]. Furthermore, a median delay of five years in diagnosing superficial endometriosis has been reported [29].

In the present study, we have shown how superficial endometriotic lesions may appear at ultrasound examination. Based on the data above reported about the evolution of superficial endometriotic lesions, our analysis related to the correlation between imaging findings and laparoscopy/histology assumed the hypothesis that initial superficial lesions are always cystic and a stromal reaction/fibrosis develops around with time, with the observation that a small pocket/retraction/peritoneal defect in laparoscopy is frequently associated, even in tiny lesions [30].

To date, literature regarding the role of transvaginal ultrasound for detecting superficial pelvic endometriosis is scarce. Some authors have reported that the assessment of some indirect sonographic signs (the so-called “soft markers”) may be associated to the presence of superficial endometriosis. 

Reid et al. analyzed data from a series of 189 women with clinical suspicion of endometriosis who underwent a detailed transvaginal ultrasound evaluation followed by laparoscopic surgery [31]. They observed that superficial endometriosis was present in 64.6% of the patients. In more than half (54.1%) of these women, superficial endometriosis was the only finding at surgery. Interestingly, the patients’ histories and complaints were not associated to the presence or absence of superficial endometriosis, indicating that clinical aspects are not useful for diagnosing superficial endometriosis. However, regarding ultrasound findings, the observation of ovarian immobility, in the absence of ovarian endometrioma and/or deep endometriosis nodules, showed a low sensitivity and positive predictive value, for both the right (7% and 14%, respectively) and left (16% and 27%, respectively) ovary in women with isolated superficial endometriosis. However, the specificity and negative predictive value was high for both ovaries (right ovary: 94% and 87%, respectively and left ovary: 87% and 78%, respectively), suggesting that visualizing a mobile ovary indicates a low probability of the presence of pelvic sidewall superficial endometriosis. These authors also observed that specific-site tenderness was associated to the presence of pelvic sidewall superficial endometriosis. It should be noted that these findings were in agreement with previous data from Yong et al. [32].

Robinson et al. use a prospective observational study evaluating 81 women with clinical suspicion of pelvic endometriosis aiming to analyze the diagnostic performance of transvaginal ultrasound for detecting superficial endometriosis in the utero-sacral ligaments [33]. Women with deep endometriosis were excluded. Surgical laparoscopic findings and/or histology was the reference standard. a total of 44 (54%) women had superficial endometriosis in one or both utero-sacral ligaments. They observed that a >5.8 mm and >6.1 mm thickness of the left and right utero-sacral ligaments, respectively, showed a 96% specificity for detecting superficial endometriosis at this anatomical site. However, sensitivity was very poor (10%). Moreover, specific-site tenderness showed a poor diagnostic performance (63% sensitivity and 43% specificity). These authors concluded that subjective thickening and site-specific tenderness of the utero-sacral ligament area do not exhibit appropriate test characteristics to be useful diagnostic or screening tools for superficial endometriosis near the utero-sacral ligaments.

Chowdary et al. analyzed the diagnostic performance of several ultrasound features for diagnosing superficial endometriosis in a retrospective study [17]. The reported data evaluated 53 women who underwent laparoscopy for pelvic pain. The variables analyzed were utero-sacral ligaments thickness, pericolic fat thickening, ovarian mobility, specific-site tenderness, and the presence of filmy adhesions. They observed that pericolic fat thickening and thickness of utero-sacral ligaments were the features most associated with the presence of superficial endometriosis. The thickness of utero-sacral ligaments had a moderate diagnostic performance (62% sensitivity and 73% specificity). No specific information was reported for the thickening of pericolic fat. In this study, ovarian mobility and filmy adhesions had no diagnostic value. This study was criticized mainly for lacking a clear definition of thickening of pericolic fat [34].

Because of the previously acknowledged limitations for detecting superficial endometriotic lesions [12,35], Leonardi et al. hypothesized that the instillation of saline through the tubes could create an acoustic window in the pelvis that could help to identify these peritoneal lesions [36]. This proposal was based on the fact that the presence of fluid in the pouch of Douglas may improve the ability to visualize anatomic structures of the posterior pelvic compartment [37]. Their proposed technique was called saline-infusion sonoPODography and consisted of performing uterine saline infusion sonography as routinely performed for detecting uterine intra-cavitary pathology or assessing tubal patency. The instilled saline flushed through the tubes and fell into the pouch of Douglas, generating an acoustic window. The recommended amount of saline to be instilled was 20 mL. The researchers analyzed the feasibility of this method in a series of 59 women. They observed that the instillation was successful in 94% of the series, and fluid in the pouch of Douglas could be seen in 73% of the women. These authors reported that the superior border of the rectovaginal septum was only visualized on sonoPODography. Similarly, successful sonoPODography outlined the tissue interface of the utero-sacral ligaments, providing a clearer view of the structure. Subjectively, the utero-sacral ligaments were more obvious and quickly seen with fluid in the pouch of Douglas. Furthermore, the interface of the pouch of Douglas peritoneum could be visualized, which was not possible in patients without fluid in the pouch of Douglas. They concluded that sonoPODography is a feasible procedure, providing a unique artificial window into the posterior compartment.

This same group has recently reported a single center prospective study assessing the diagnostic performance of sonoPODography [18]. The series included 42 women who underwent sonoPODography as an index test and laparoscopic and/or histologic confirmation of superficial endometriosis as a reference test. Superficial endometriosis was diagnosed, as per the reference test, in 37 women (88%). The diagnosis of superficial endometriosis was based on the observation of at least one of the following findings: (1) hyperechoic projections, (2) hypoechoic areas, (3) filmy adhesions, (4) cystic areas, and (5) peritoneal pockets, as demonstrated by incomplete septations and “entrapped” fluid. Interestingly, the possible appearances of superficial endometriosis were defined relative to the peritoneal surface with <5 mm of depth. The sensitivity and specificity of the technique were 65% and 100%, respectively. Albeit very interesting, minimal invasion is required for this method. Furthermore, this technique only allows for a better visualization of lesions located in the peritoneum of the pouch of Douglas, the utero-sacral ligaments, the posterior pelvic sidewall, and the rectosigmoid, but it probably does not improve the ability for observing the peritoneum of the bladder and vesicouterine pouch.

In our study, we attempted to define the ultrasound features that may exhibit superficial endometriotic lesions in laparoscopically confirmed lesions. Our findings need to be confirmed in future prospective studies. Nevertheless, this proposal could be the basis for a consensus about how to describe these lesions. In addition, the reproducibility of these descriptions should be assessed in future studies.

The main strength of our study is that, to the best of our knowledge, this is the first proposal for describing the ultrasound features of superficial endometriotic lesions. However, we are aware of some study limitations, such as the small series we are reporting and the absence of statistical analysis.

## 5. Conclusions

Our study shows that superficial endometriosis lesions can be observed by transvaginal ultrasound. The lesions can be described and exhibit different ultrasound features. Future research is needed to evaluate the diagnostic accuracy of this technique and to analyze the correlation between the ultrasound features, the type of lesions observed in laparoscopy, and the clinical complaints.

## Figures and Tables

**Figure 1 diagnostics-13-01876-f001:**
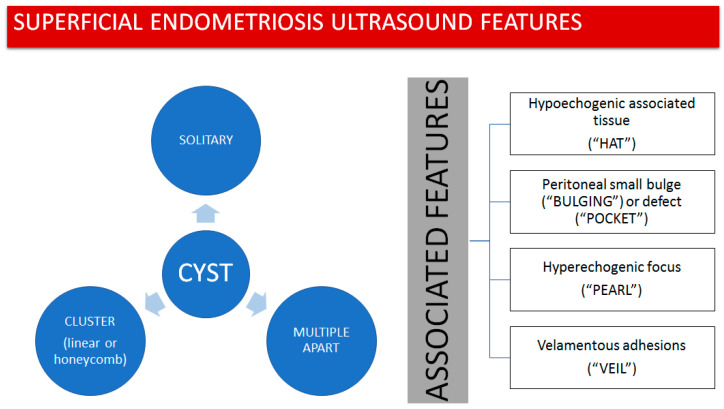
Proposed classification for superficial endometriotic lesions.

**Figure 2 diagnostics-13-01876-f002:**
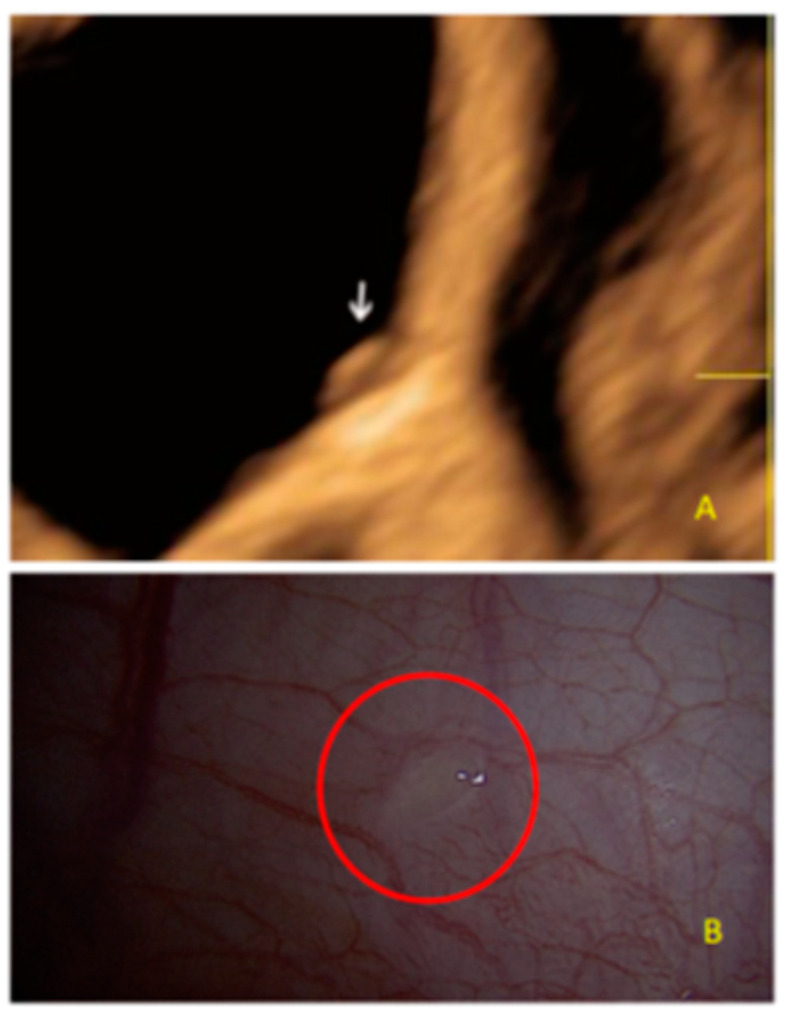
Superficial endometriotic solitary cystic lesion ((**A**), arrow) with the corresponding laparoscopic findings (**B**), red circle shows lesion as seen by laparoscopy.

**Figure 3 diagnostics-13-01876-f003:**
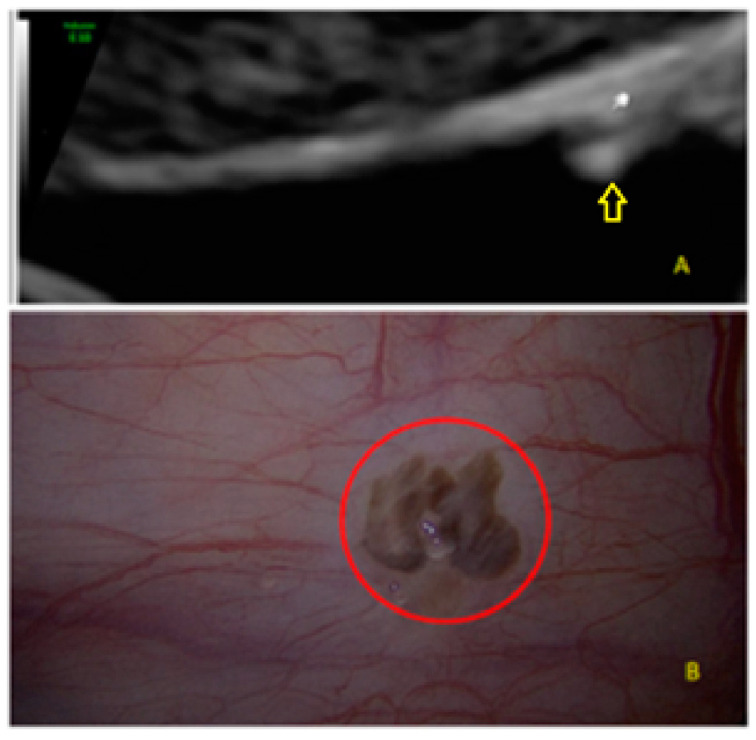
Superficial endometriotic solitary bulging lesion (**A**, arrow) with the corresponding laparoscopic findings (**B**), red circle shows lesion as seen by laparoscopy.

**Figure 4 diagnostics-13-01876-f004:**
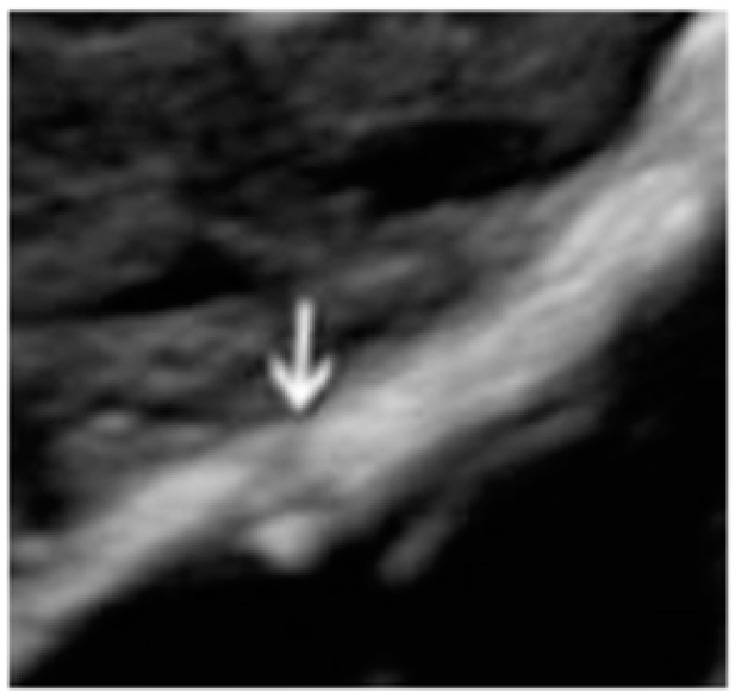
Superficial endometriotic solitary bulging lesion (arrow).

**Figure 5 diagnostics-13-01876-f005:**
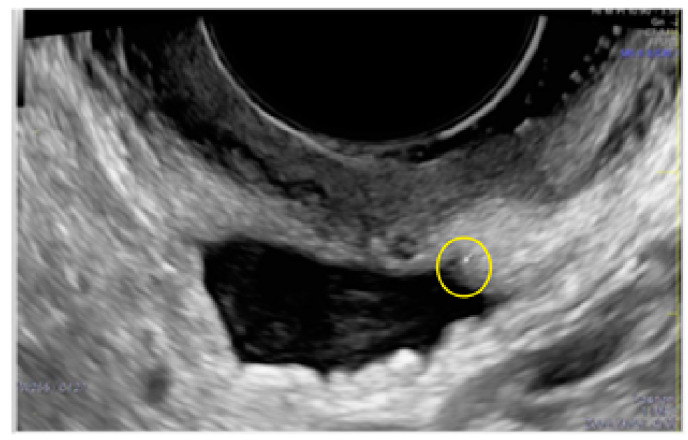
Superficial endometriotic solitary pocket lesion (circle, arrow).

**Figure 6 diagnostics-13-01876-f006:**
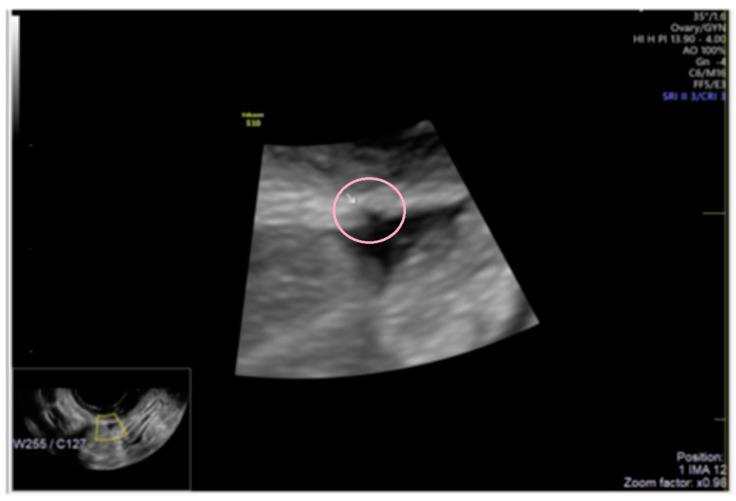
Superficial endometriotic solitary pocket lesion with a “hat” (circle and arrow).

**Figure 7 diagnostics-13-01876-f007:**
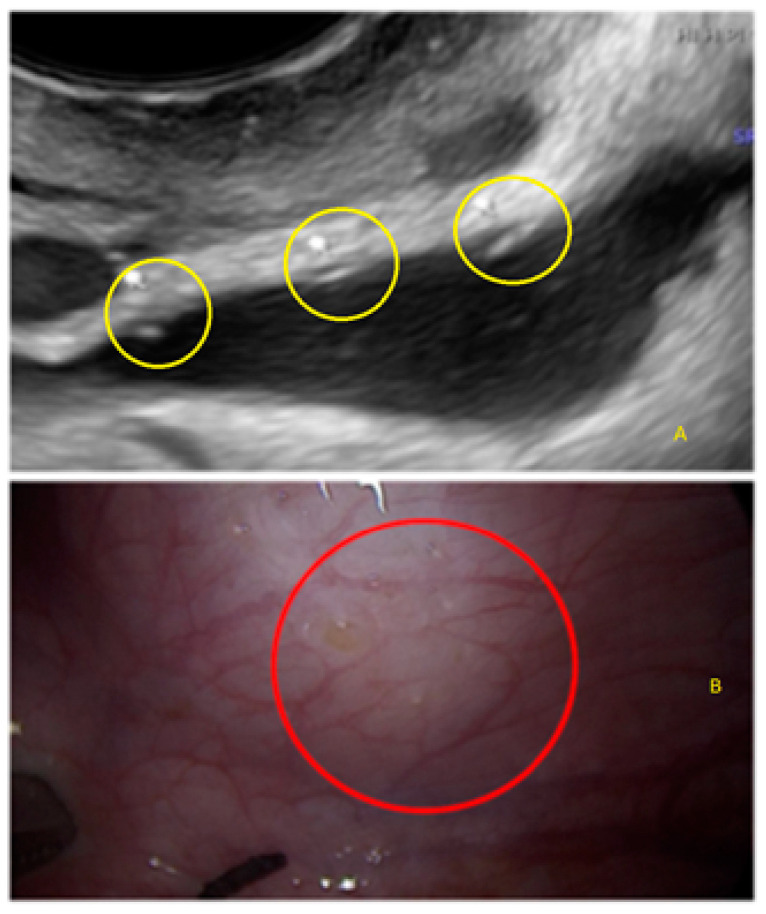
Superficial endometriotic multiple separate bulging lesions ((**A**), yellow circles), with the corresponding laparoscopic findings ((**B**), red circle).

**Figure 8 diagnostics-13-01876-f008:**
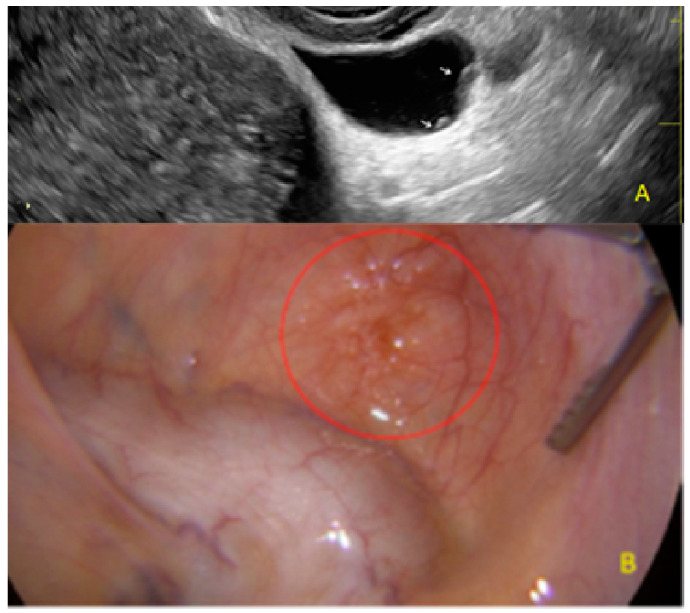
Superficial endometriotic multiple separate bulging lesions with hypoechoic tissue ((**A**) arrows), with the corresponding laparoscopic findings ((**B**), red circle).

**Figure 9 diagnostics-13-01876-f009:**
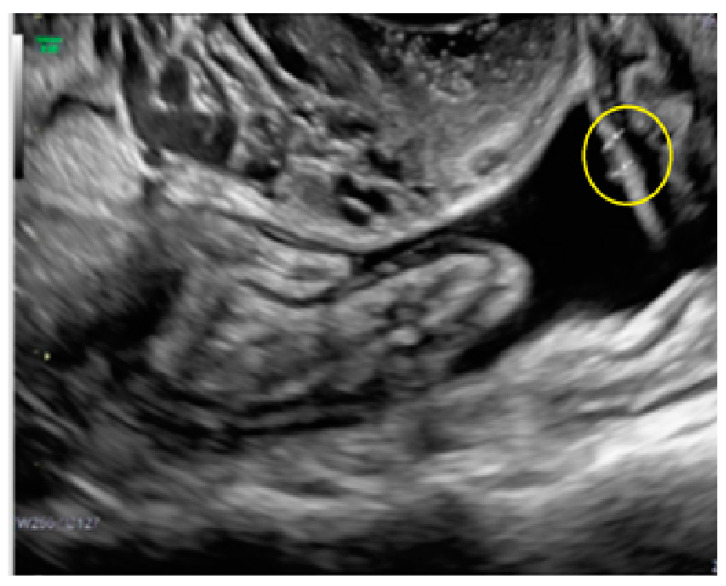
Superficial endometriotic multiple separate slightly bulging lesions with hyperechoic foci (circle, arrow).

**Figure 10 diagnostics-13-01876-f010:**
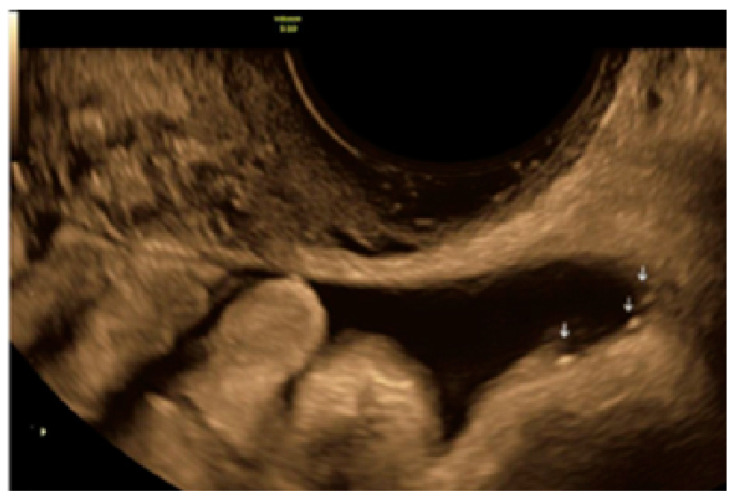
Other superficial endometriotic multiple separate slightly bulging lesions with hyperechoic foci (arrows).

**Figure 11 diagnostics-13-01876-f011:**
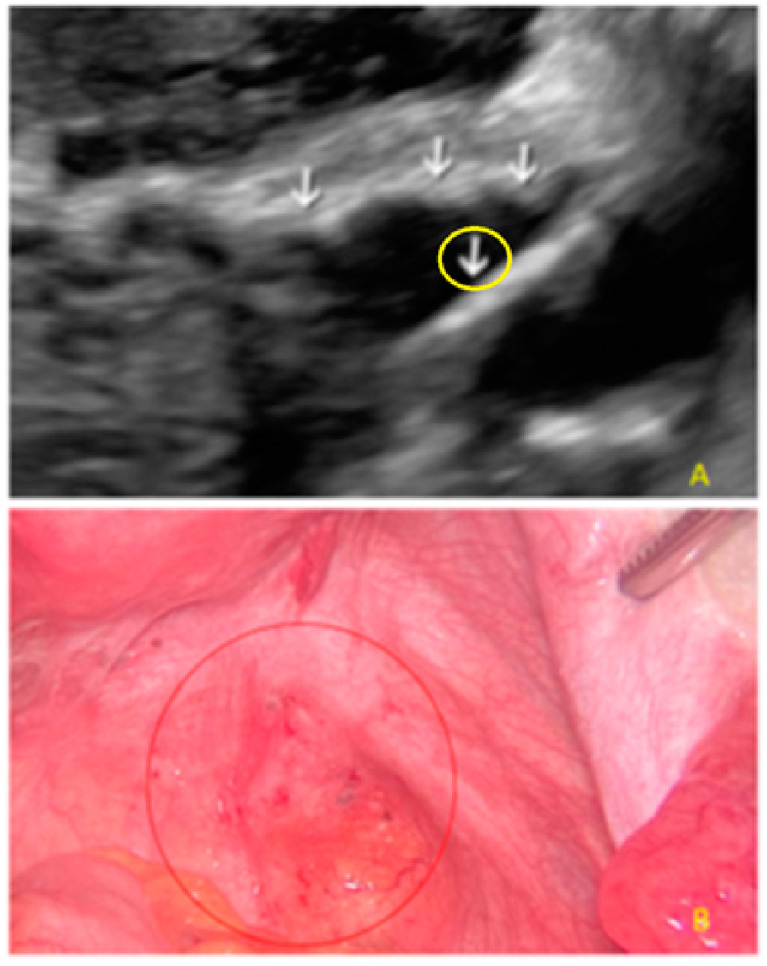
Superficial endometriotic multiple separate bulging lesions ((**A**), arrows) with velamentous adhesion ((**A**), arrow with yellow circle), and without hyperechoic foci, with the corresponding laparoscopic findings ((**B**), red circle).

**Figure 12 diagnostics-13-01876-f012:**
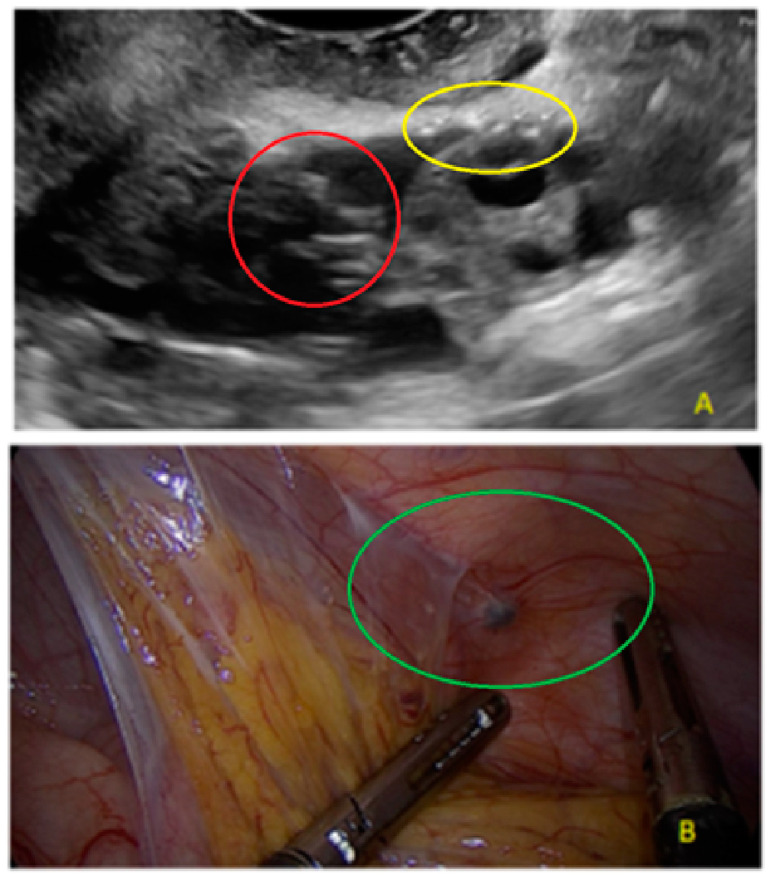
Superficial endometriotic multiple separate slightly bulging lesions with hyperechoic foci (pearl) ((**A**), yellow circle) and velamentous adhesion (veil) ((**A**), red circle), with the corresponding laparoscopic findings ((**B**), green circle).

**Figure 13 diagnostics-13-01876-f013:**
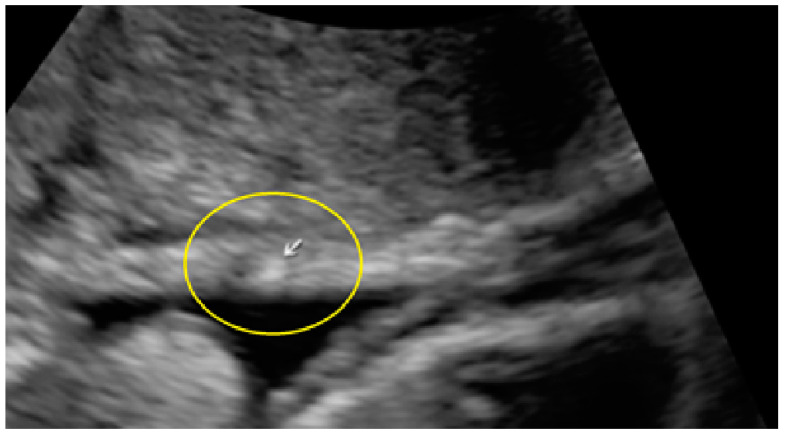
Superficial endometriotic lesion with hypoechogenic surrounding tissue (yellow circle, arrow).

**Figure 14 diagnostics-13-01876-f014:**
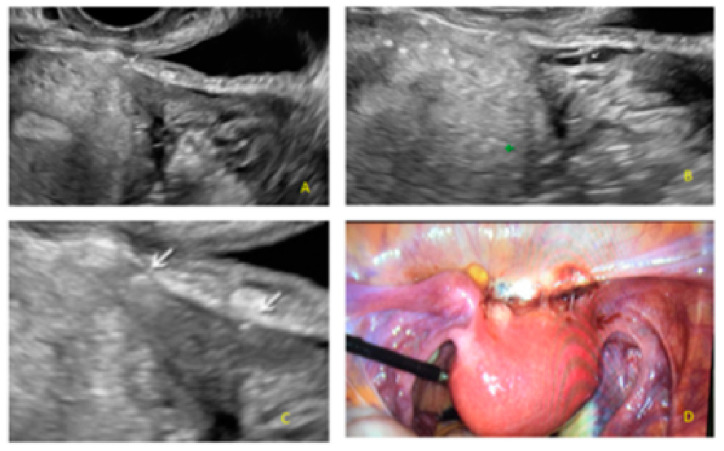
Superficial endometriotic multiple separate slightly bulging lesions with hyperechoic foci (pearl) (**A**,**C**, arrows) with velamentous adhesions (veil) (**B**, arrow) and the corresponding laparoscopic findings (**D**).

**Figure 15 diagnostics-13-01876-f015:**
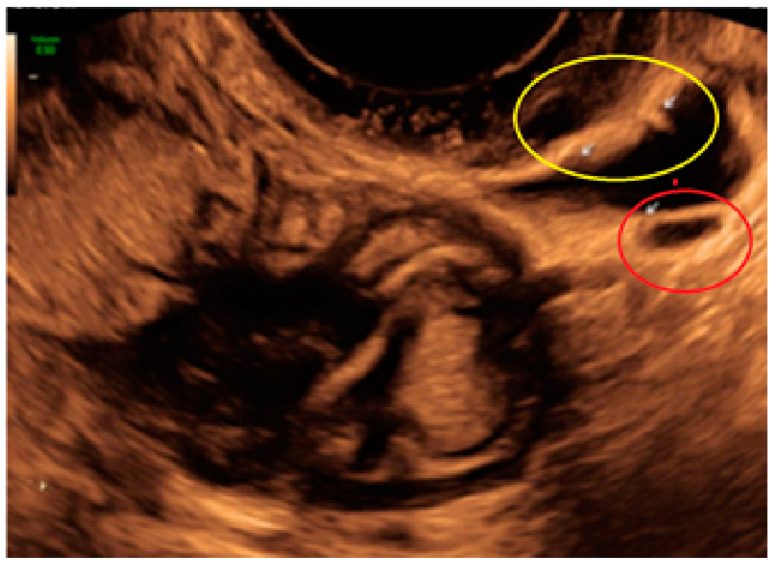
Superficial endometriotic multiple separate slightly bulging lesions with hyperechoic foci (pearl) (yellow circle, arrows) with velamentous adhesions (veil) (red circle, arrow).

**Figure 16 diagnostics-13-01876-f016:**
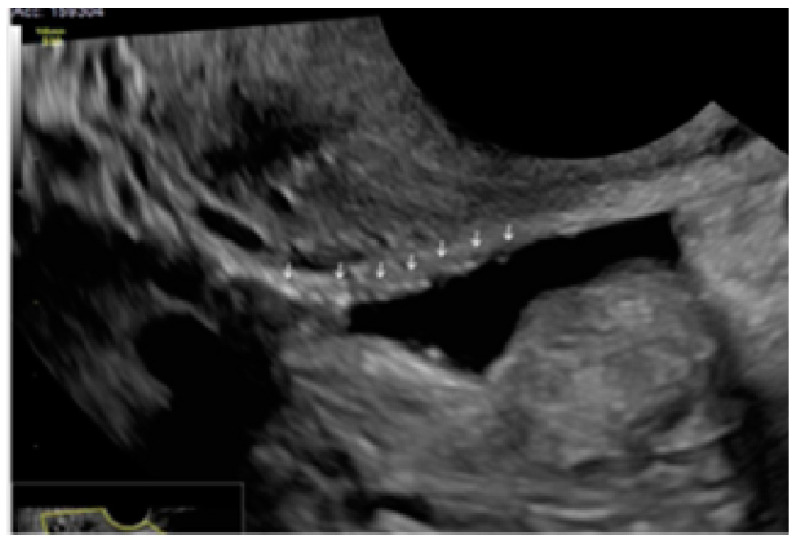
Superficial endometriotic lesions disposed on a linear cluster (arrows).

**Figure 17 diagnostics-13-01876-f017:**
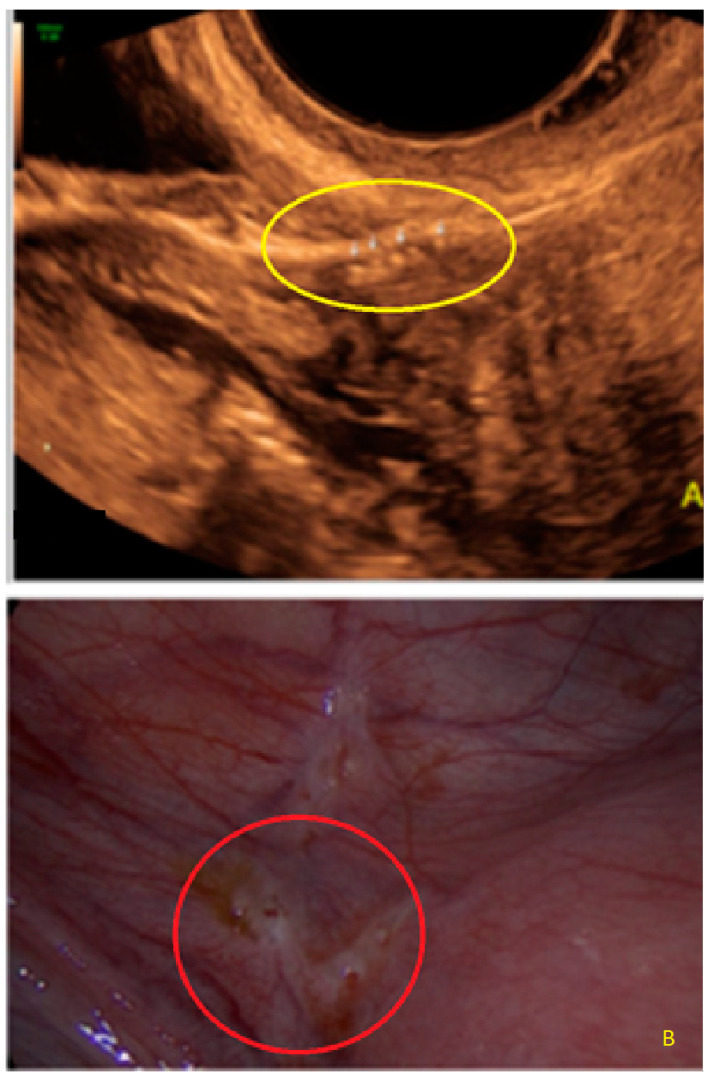
Superficial endometriotic lesions in a linear cluster with hyperechoic foci (pearl) ((**A**), yellow circle, arrows) with laparoscopic findings ((**B**), red circle).

**Figure 18 diagnostics-13-01876-f018:**
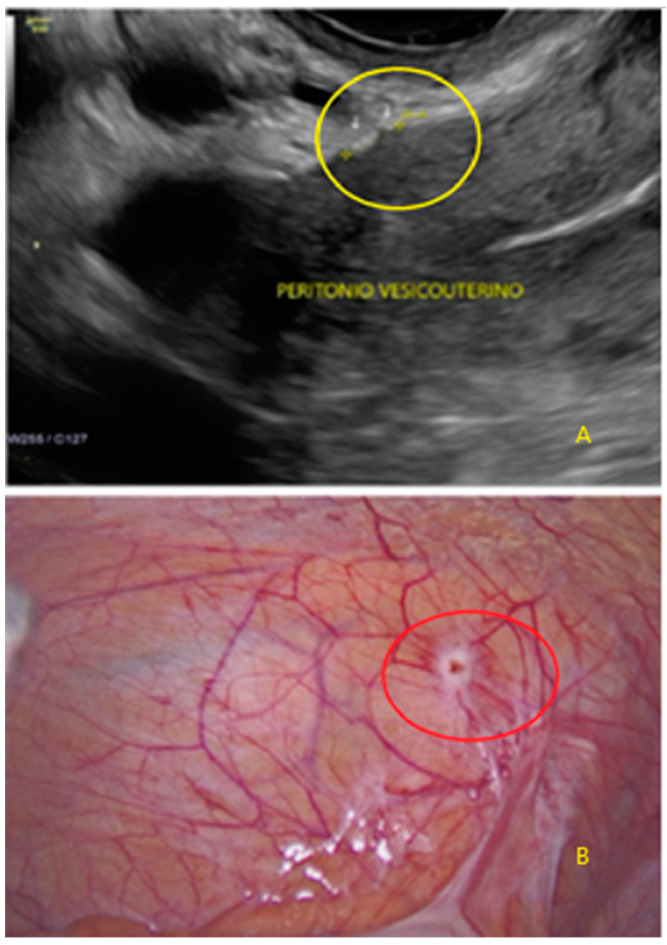
Superficial endometriotic lesions in a linear cluster with hyperechoic foci (pearl) ((**A**), yellow circle, arrows) with laparoscopic findings ((**B**), red circle), located in the peritoneum of the vesicouterine pouch.

**Figure 19 diagnostics-13-01876-f019:**
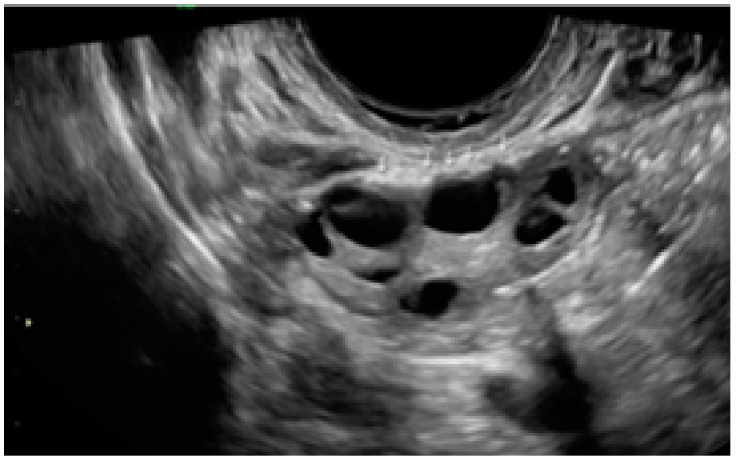
Another example of superficial endometriotic lesions in a linear cluster with hyperechoic foci, in this case over ovarian surface (pearl) (arrows).

**Figure 20 diagnostics-13-01876-f020:**
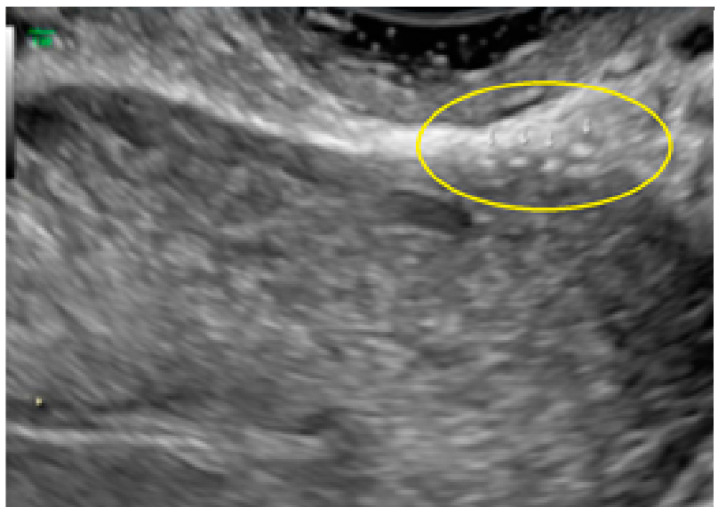
Superficial endometriotic lesions in a linear cluster with hyperechoic foci (pearl) with hypoechoic tissue (hat) (yellow circle, arrows), located in the peritoneum of the vesico-uterine pouch.

**Figure 21 diagnostics-13-01876-f021:**
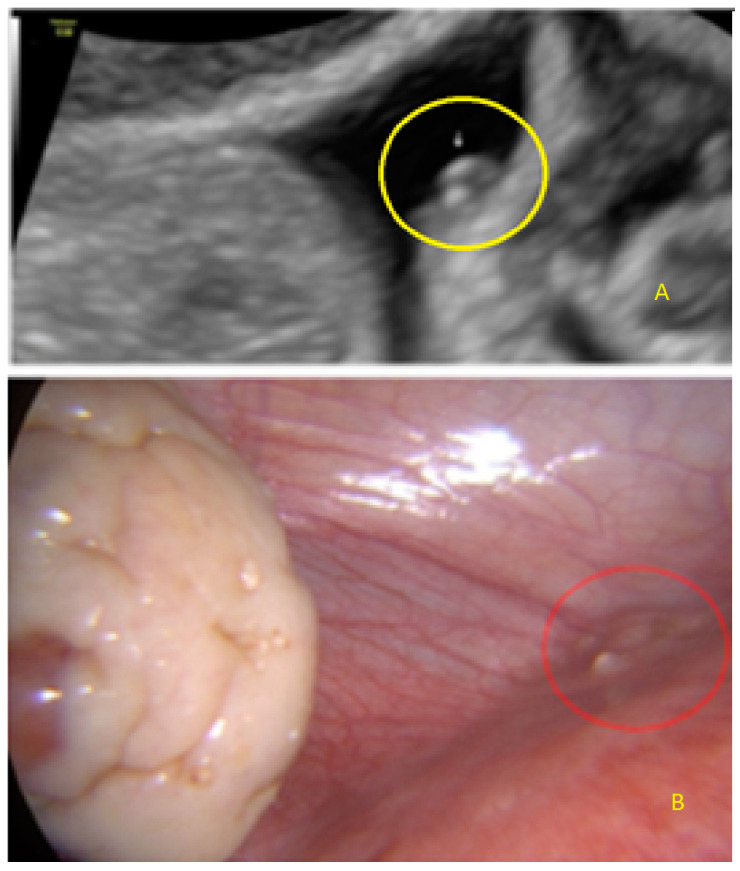
Superficial endometriotic lesions in cluster with honeycomb appearance, convex to the peritoneal surface (bulging) ((**A**), yellow circle, arrow) with laparoscopic findings ((**B**), red circle).

**Figure 22 diagnostics-13-01876-f022:**
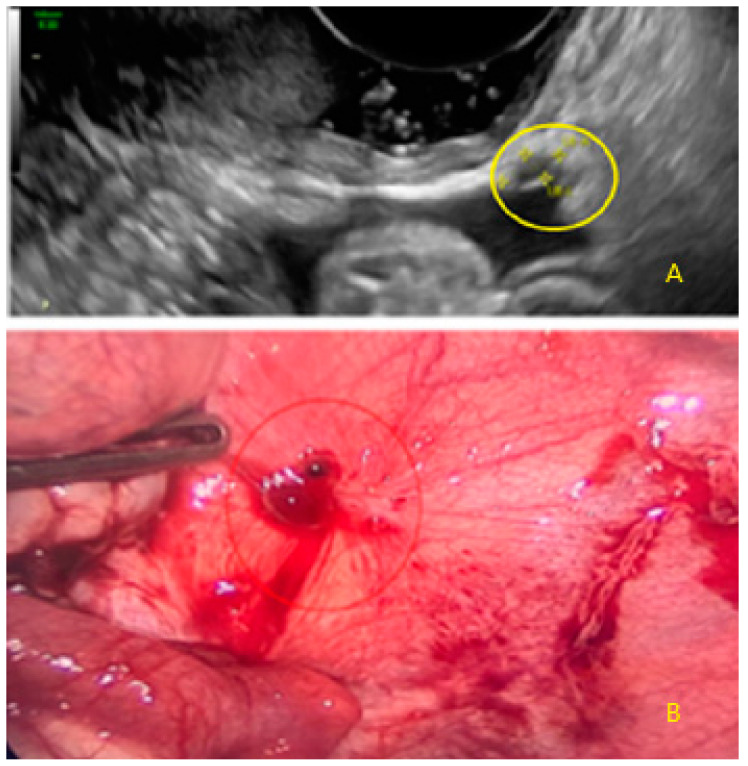
Another example of superficial endometriotic lesions in cluster with honeycomb appearance, convex to the peritoneal surface (bulging) ((**A**), yellow circle, callipers) with laparoscopic findings ((**B**), red circle).

**Figure 23 diagnostics-13-01876-f023:**
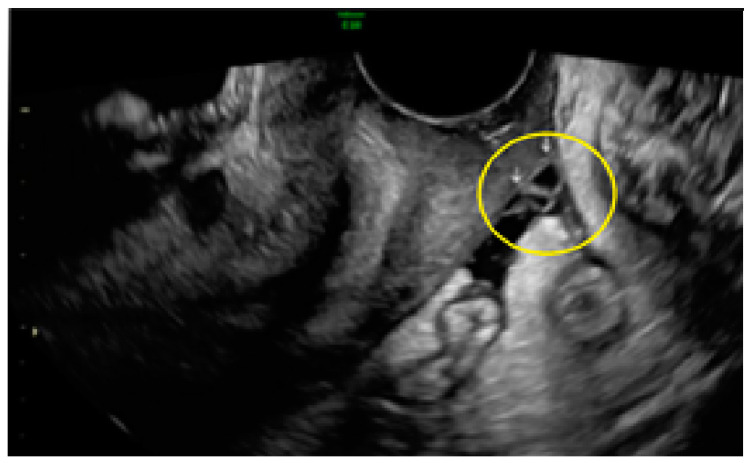
Example of superficial endometriotic lesions in cluster with honeycomb appearance, convex to the peritoneal surface (bulging) (yellow circle, arrows) located in the pouch of Douglas.

**Figure 24 diagnostics-13-01876-f024:**
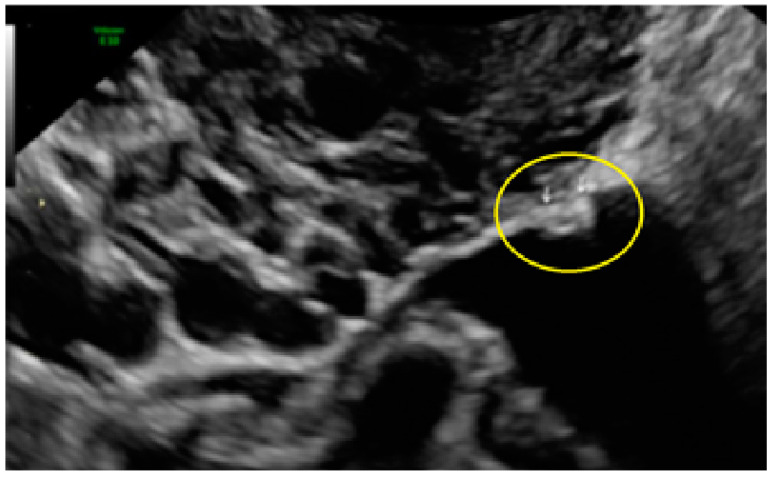
Superficial endometriotic lesions in cluster with honeycomb appearance, convex to the peritoneal surface (bulging) (yellow circle, arrows) located in the surface of the uterosacral ligament.

**Figure 25 diagnostics-13-01876-f025:**
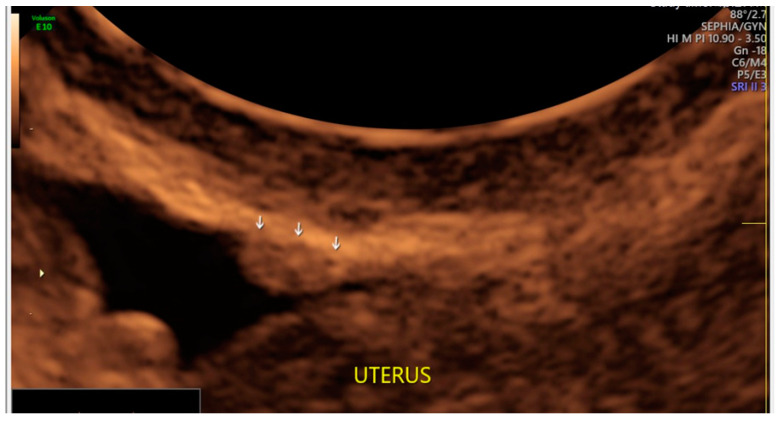
Another example of superficial endometriotic lesions in cluster with honeycomb appearance, convex to the peritoneal surface (bulging) (Arrows) located in the surface of the uterosacral ligament.

**Figure 26 diagnostics-13-01876-f026:**
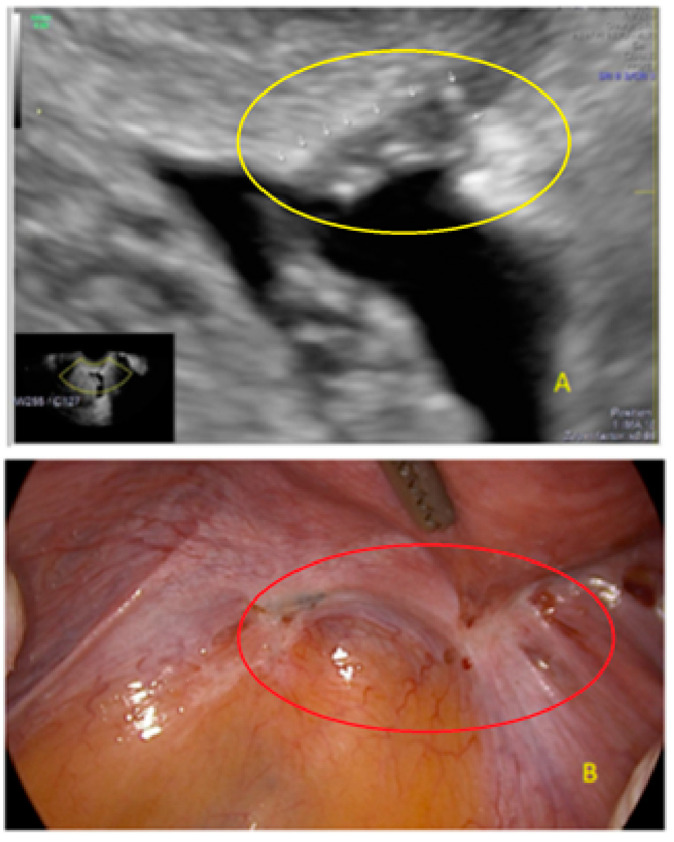
Superficial endometriotic lesions in cluster with honeycomb appearance, convex to the peritoneal surface (bulging) with hypoechoic tissue surrounding the lesion ( hat) ((**A**), yellow circle, arrows), with laparoscopy ((**B**), red circle).

**Figure 27 diagnostics-13-01876-f027:**
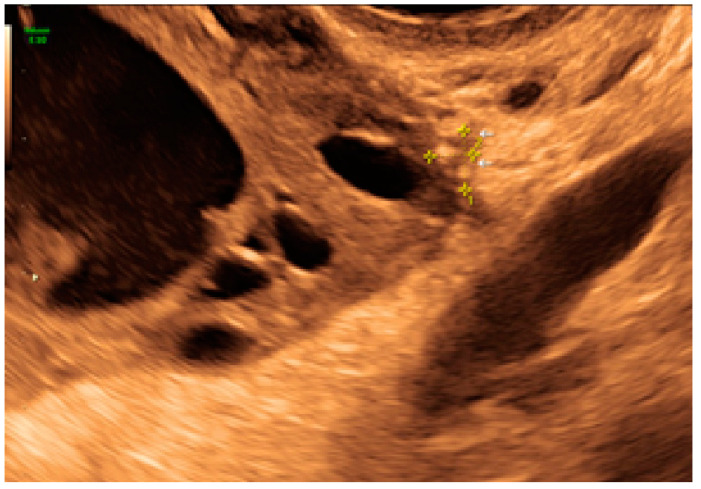
Superficial endometriotic lesions in cluster with honeycomb appearance, convex to the peritoneal surface (bulging) with hyperechoic foci (pearl) (calipers, arrows).

**Figure 28 diagnostics-13-01876-f028:**
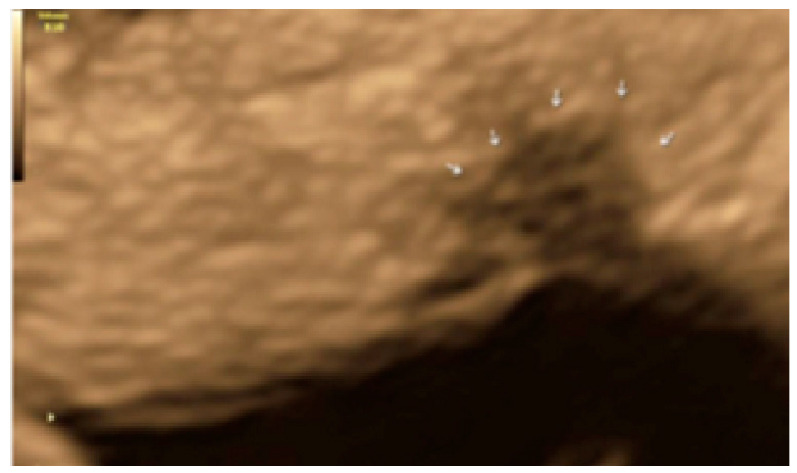
Superficial endometriotic lesions in a cluster with a honeycomb appearance, concave to the peritoneal surface (pocket) (arrows).

**Figure 29 diagnostics-13-01876-f029:**
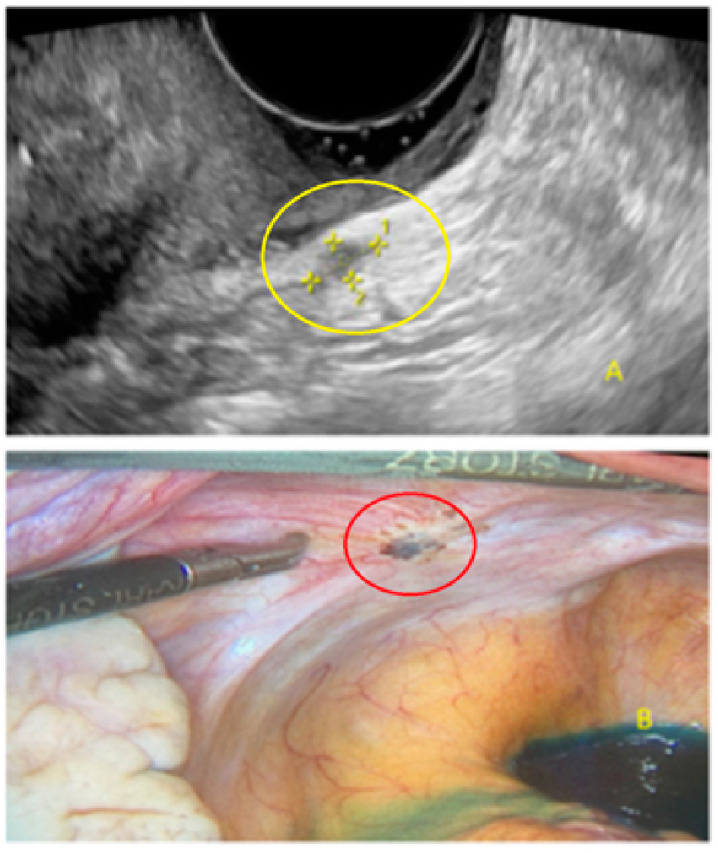
Superficial endometriotic lesions in a cluster with a honeycomb appearance, concave to the peritoneal surface (pocket) ((**A**), yellow circle, callipers) with laparoscopic findings ((**B**), red circle).

**Figure 30 diagnostics-13-01876-f030:**
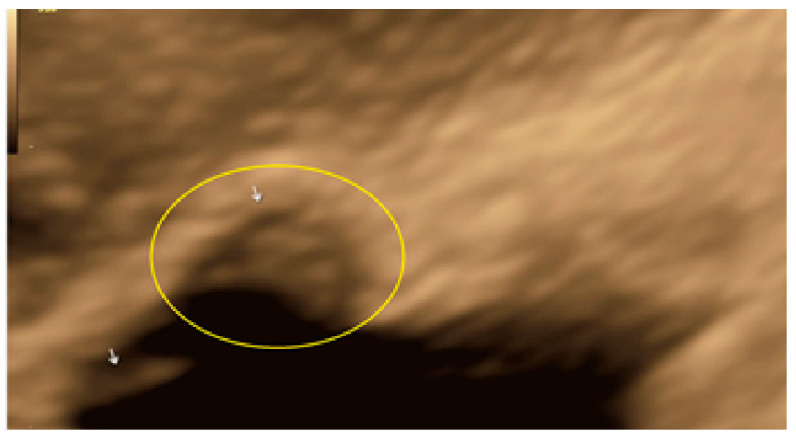
Another example of superficial endometriotic lesions in cluster with honeycomb appearance, concave to the peritoneal surface (pocket) (yellow circle, arrows).

**Figure 31 diagnostics-13-01876-f031:**
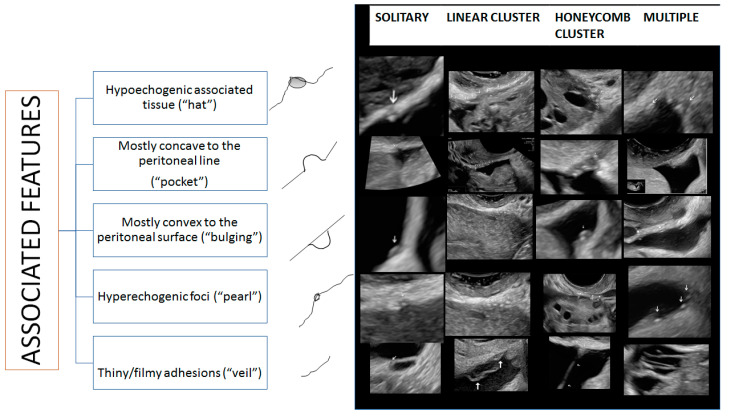
Schematic representation of superficial endometriosis lesions. Lesions marked by arrows.

## Data Availability

Data are available upon reasonable request.

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
