# Peer review of "Superficial Endometriosis at Ultrasound Examination—A Diagnostic Criteria Proposal"

_diagnostics, 2023, doi:10.3390/diagnostics13111876_

Round 1

Reviewer 1 Report

Dear Author,

Is an interesting article that brings useful information in the diagnosis of endometriosis.

The quality of the figures and tables is satisfactory.

The reference list covers the relevant literature adequately and impartially.

Statistical methods are valid and correctly applied.

In my opinion  it meets the conditions for publication.

Kind regards,

Author Response

  1. Question: Is an interesting article that brings useful information in the diagnosis of endometriosis. The quality of the figures and tables is satisfactory. The reference list covers the relevant literature adequately and impartially. Statistical methods are valid and correctly applied. In my opinion  it meets the conditions for publication. Kind regards,
  2. Answer: Thanks for these comments. We appreciate them. No change made on the manuscript

Reviewer 2 Report

1.      I truly stand that this study is a valuable one, it defines and describes the ultrasound features of superficial endometriotic lesions, usually confirmed by the laparoscopy. It’s crucial all of us to use the same terminology. The diagnosis of the peritoneal endometriosis, in an early stage, is an important issue in gynaecology and to detect it, the study uses a non-invasive method. 

2.      I have some doubts whether the description of these sonographic features is reproducible or not. I think that is possible in solitary, multiple apart and cluster classification groups, but more difficult in hat, pocket, pearl and veil. I hope it will be the target of further studies.

3.      In the conclusions the authors affirm that this study shows that superficial endometriosis can be diagnosed by transvaginal ultrasound. If this means that some cases of superficial endometriosis are diagnosable by transvaginal ultrasound I completely agree, but I think that this study, due to its design, cannot evaluate the sensibility of the method. This could be achieved in my opinion by analysing a group of symptomatic women having a laparoscopic diagnosis of endometriosis (currently the gold standard) and comparing the number of the ones with a previous positive US for endometriosis with the number of patients with a negative one. Furthermore, the study should declare if all the lesions found thanks to the laparoscopy were described before by the transvaginal ultrasound.

4.      The references are relevant to the research, but I am not sure all if all of them deserve the same highlight.

5.      There are some mistakes:

a-     the reference 38 is not cited in the text.

b-     Row 203: arrows not circle

c-      Rows 215-219: the description in the text does not correspond at the one of the figure 13

d-     Row 268: in the figure 25 there is not a yellow circle.

e-     Row 278: in the figure 27 there is not a yellow circle

Author Response

  1. Comment: I truly stand that this study is a valuable one, it defines and describes the ultrasound features of superficial endometriotic lesions, usually confirmed by the laparoscopy. It’s crucial all of us to use the same terminology. The diagnosis of the peritoneal endometriosis, in an early stage, is an important issue in gynaecology and to detect it, the study uses a non-invasive method. 
    1. Answer: Thanks for these comment. We appreciate
  2. Comment:  I have some doubts whether the description of these sonographic features is reproducible or not. I think that is possible in solitary, multiple apart and cluster classification groups, but more difficult in hat, pocket, pearl and veil. I hope it will be the target of further studies.
    1. Answer: Thanks for this comment. We added acomment about this point in the Discussion
  3. Comment:In the conclusions the authors affirm that this study shows that superficial endometriosis can be diagnosed by transvaginal ultrasound. If this means that some cases of superficial endometriosis are diagnosable by transvaginal ultrasound I completely agree, but I think that this study, due to its design, cannot evaluate the sensitivity of the method. This could be achieved in my opinion by analysing a group of symptomatic women having a laparoscopic diagnosis of endometriosis (currently the gold standard) and comparing the number of the ones with a previous positive US for endometriosis with the number of patients with a negative one. Furthermore, the study should declare if all the lesions found thanks to the laparoscopy were described before by the transvaginal ultrasound.
    1. Answer: We fully agree. In fact, the stated in the Conclusion that future research is needed for assessing the diagnostic accuracy of TVS for detecting SE lesions .A slight amendment have been made in the Discussion
  4. Comment: The references are relevant to the research, but I am not sure all if all of them deserve the same highlight.
    1. Thansk for this comment. We do think all references are relevant for this paper
  5. Comment: the reference 38 is not cited in the text.
    1. Answer: Sorry for this mistake. Referecne is cited in the Revised Version
  6. Comment: Row 203: arrows not circle
    1. Answer: Amended
  7. Comment: Rows 215-219: the description in the text does not correspond at the one of the figure 13
    1. Answer: Amended
  8. Comment: Row 268: in the figure 25 there is not a yellow circle.
    1. Answer: Amended
  9. Comment: Row 278: in the figure 27 there is not a yellow circle
    1. Answer: Amended

Reviewer 3 Report

Superficial endometriosis is a relevant clinical problem because of its prevalence and diagnostic
difficulties.
The literature about the role of transvaginal ultrasound for detecting pelvic superficial endometriosis is scanty.
And there are only some suggestions about indirect sonographic signs called “soft markers”.
This paper is a pictorial assay where the authors show how superficial endometriotic lesions may appear at
ultrasound examination.
The contribution to the scientific research is high , in this field.
An important strength is the exact correlation with the laparoscopic representation which makes the description
of the ultrasound findings very effective.
The manuscript is clear and presented in a well structured manner.
The scientific design is appropriate as weel as methods section.
From a personal point of view the distinction between “bulging” and “hat “is a little less clear
However the pictures are exemplary.
The conclusions are consistent
The literature, not self-referenced, is very up-to-date and the review reported under discussion is very good.

Author Response

  1. Comment: Superficial endometriosis is a relevant clinical problem because of its prevalence and diagnostic difficulties. The literature about the role of transvaginal ultrasound for detecting pelvic superficial endometriosis is scanty.
    And there are only some suggestions about indirect sonographic signs called “soft markers”. This paper is a pictorial assay where the authors show how superficial endometriotic lesions may appear at ultrasound examination. The contribution to the scientific research is high in this field. An important strength is the exact correlation with the laparoscopic representation which makes the description of the ultrasound findings very effective. The manuscript is clear and presented in a well structured manner. The scientific design is appropriate as weel as methods section. From a personal point of view the distinction between “bulging” and “hat “is a little less clear However the pictures are exemplary. The conclusions are consistent. The literature, not self-referenced, is very up-to-date and the review reported under discussion is very good.
  2. Answer: Thanks for these comments. We appreciate them. The main difference between "HAT" and "Bulging" is that the lesion does not protrude the peritoneum. Concept clarified in the revised version. Highlighted in yellow.